# The Meaning of Healing to Adult Patients with Advanced Cancer

**DOI:** 10.3390/ijerph20021474

**Published:** 2023-01-13

**Authors:** Eve Namisango, Emmanuel B. K. Luyirika, Lawrence Matovu, Ann Berger

**Affiliations:** 1African Palliative Care Association, Kampala P.O. Box 72518, Uganda; 2Department of Palliative Care and Rehabilitation, Cicely Saunders Institute King’s College London, London SE5 9RS, UK; 3Formerly-Department of Clinical Services, Hospice Africa Uganda, Kampala P.O. Box 7757, Uganda; 4National Institutes of Health, Clinical Center, Bethesda, MD 20892, USA

**Keywords:** meaning, healing, spirituality, advanced cancer, sub-Saharan Africa

## Abstract

Background: This study aimed to explore the meaning of healing from the perspective of adult patients with advanced cancer. Methods: We conducted a secondary analysis of data from a primary study which used a cognitive interview approach to assess the face and content validity of a spiritual and psychological healing measure (NIH-HEALS). This analysis focused on responses to the question, “What does the term ‘healing’ mean to you?” Data were de-identified, transcribed verbatim, and imported in NVivo for thematic analysis in line with interpretive phenomenological methods. Results: Thirty-five adults with advanced cancer participated in the study. We identified nine major themes: acceptance, surrender, faith, hope, peace, freedom from suffering (e.g., pain, problems, or other bothersome factors), overcoming/transcending disease, positive emotions (e.g., happiness), recovery from illness or disease. One participant discussed healing as synonymous with death, and two associated it with social relations and social support. Conclusion: Themes from patients’ responses suggest subjective and varied definitions of healing which encompass physical, social, spiritual, and psychological domains of well-being, distinct from the physical cure of disease. Clinicians should adopt a holistic, person-centered approach to care, attending to bodily, psychosocial, spiritual, and emotional needs to help patients find meaning in their experiences, nourish resilience, and experience a sense of healing—as they define it.

## 1. Introduction

Cancer remains a major global public health concern accounting for over 10 million deaths in 2020 [1]. The burden of cancer is projected to increase by one million deaths per year by 2030, with most cases occurring in low-middle income settings [2]. Notably, most of the patients in resource-limited settings present late with advanced disease with limited options for cure and have high needs for palliative care. 

Cancer is associated with complex symptoms and concerns which are very distressing [3] and negatively impact the quality of life of the patient and their family. These symptoms and concerns include physical, social, psychological, and spiritual domains; thus, they have a multi-dimensional impact on patient well-being and their experience of suffering. Common physical symptoms include fatigue, pain [4], lack of energy, weakness and appetite loss [3,5], psychological symptoms include anxiety, depression/depressive symptoms (such as low mood, sadness), nervousness, and worry [5]. Some of the commonly reported spiritual symptoms include lack of peace, spiritual pain and examples of social concerns include information, social support, and psychosocial distress [5]. 

Palliative care aims to alleviate symptoms and concerns that cause serious health-related suffering. Palliative care provision is recognized as a core component of comprehensive cancer care, universal health coverage, and the highest standards of physical and mental health, which should be made equitably available across the disease trajectory–starting at the time of cancer diagnosis and continuing to end-of-life and bereavement [6,7]. It uses a holistic care approach to meet the needs of seriously ill patients and their families, knowing that these go beyond the physical symptoms and lead to suffering and potential loss of dignity as the intactness of patients are threatened [8]. The palliative care fraternity further notes that “Palliative care is the active holistic care of individuals across all ages with serious health-related suffering due to severe illness and especially of those near the end of life. It aims to improve the quality of life of patients, their families and their caregivers.” [9]. 

The World Health Organisation also emphasizes the need for both people-centered and person-centered models of care. “Person-centered models of care an approach to care that consciously adopts individuals’, carers’, families’ and communities’ perspectives as participants in, and beneficiaries of, trusted health systems that respond to their needs and preferences in humane and holistic ways.” [10]. The approach also emphasizes people empowerment and engagement in shaping the care that they receive. Person-centered models of care see the person as a whole who also has different levels of needs and goals which are driven by personal determinants of health [10]. In this way, we can ensure that the care mirrors patient and family priorities. The United Nations political declaration of the high-level meeting on Universal Health Coverage affirms that every human being has a right to attain the highest standard of physical and mental health [11], people and person-centered models of care further advance this affirmation 

The International Alliance of Patients’ Organisations defines person-centered (or patient-centered) care (PCC) as “focused and organised around people, rather than disease” [12]. PCC views individuals, families and communities as participants in health systems responsive to their needs, PCC takes a holistic approach, focusing on the **individual** rather than solely on disease. To deliver person-centered care, health workers must be able to discuss goals of care for patients with advanced cancer to empower them to make informed treatment decisions. That said, some health workers in sub-Saharan Africa do not commonly discuss prognosis with their patients and as such many patients are often unaware of the their treatment goals [13]. Evidence shows a bias towards focusing on cure as a goal of care [14], and within western biomedicine, the concept of healing is often synonymous with curing disease or restoring health for the medically ill [15]. When cure is no longer an option as is with the case of advanced disease, some care providers may believe there is no further care available and patients may feel as if they’ve no other option but to await death. However, a recently growing body of literature demonstrates that even when cure is not possible, patients can experience healing which occurs when patients experience positive social and psychological changes in their lives irrespective of the disease-related outcome [16]. Experienced hospice and palliative care professionals have highlighted that healing is multi-dimensional, subjectively experienced differently by individuals, and spans physical, social, spiritual, and psychological domains of well-being [17]. 

Spiritual well-being is rather under researched and underdeveloped, despite its importance. Spiritual well-being emphasizes the sense of meaning in life, harmony, peacefulness, a sense of strength, and comfort from one’s faith [18]. Previous research has demonstrated the importance of aspects such as existential/spiritual well-being in such patients and that this domain has the highest potential for growth amidst declining functionality and advancing disease [10]. A study which aimed to examine the effect of spirituality on health-related quality of life in men with prostate cancer showed spirituality was associated with changes in psychosocial and physical aspects of health related quality of life [19]. Notably, people who consider themselves religious/spiritual may experience changes in their religiosity and/or spirituality depending on how they interpret the event of the terminal diagnosis. Some may interpret this as a punishment from God, or a calling to suffer for the healing of others and hence find a compelling reason to fight on. More spirituality is associated with positive changes in physical and psychosocial in health-related quality of life [19]. Positive religious coping has also been associated with better quality of life, existential well-being and positive social support in adult patients with advanced cancer [20]. A recent systematic review on spirituality in serious illness emphasized the importance of spiritual community and spiritual care as integral to health outcomes as a part of person-centered and value-sensitive care [21]. 

A meta-analysis which included over 32,000 adult cancer patients showed that positive religion and spirituality were associated with overall improved patient physical health, physical well-being, functioning well-being [22]. This draws our attention to the important observation that patients with advanced disease can draw more on their inner resources for strength to transform the suffering into positive spiritual and psychological experiences to remain resilient. This partly explains why some patients experience a high sense of well-being as they die [23]. This links into the notable and important construct of spiritual and psychosocial healing, defined as an interactive process that emerges when a person within a given disease context seeks progression towards desired wholeness [24]. It has been described as a natural, active and multi-dimensional process that is individually expressed with common patterns [25]. 

Understanding the meaning of “healing” from the perspective of patients is thus pivotal to defining goals of care, defined as “overarching aims of medical care for a patient that is informed by patient’s underlying values and priorities established within the existing clinical context” [26]. This study aimed to examine what the term “healing” means to patients with advanced cancer; to advance the state of evidence regarding the conceptualization of the construct of healing from the patients’ perspective and to stimulate the development of person-centered services which mirror what matters to patients. 

## 2. Materials and Methods

This is a secondary analysis of qualitative data from a larger cross-sectional qualitative study that used the cognitive interviewing methodology. Cognitive interviewing is an open-ended elicitation method used to understand respondents’ reasoning process for answering closed-ended statements, discerning what they interpret the statement to mean, and identifying any difficulties they may experience while answering the questions [27]. This “think aloud” interview method is useful for identifying potentially confusing or complicated words and phrases in a given statement or phrase and whether the meaning intent is well articulated by the population for whom the measure is meant for [28]. It also helps identify any missing aspects of the cognitive domain not covered by the statements. The primary study aimed to assess the face and content validity of the National Institute of Health Healing Experience during All Life Stressors (NIH-HEALS) in the population of patients with advanced cancer in Uganda and to culturally adapt this measure. 

The NIH-HEALS assessment includes questionnaire items relevant to both psycho-social, as well as spiritual healing originally developed in the United States by the NIH Clinical Center Pain and Palliative Care Service [29] and has been assessed for face and content validity in Uganda [30]. It assesses positive transformation in response to challenging life events such as cancer diagnosis. The HEALS is a self-report measure consisting of 35 items and is scored on a five-point Likert Scale from “strongly disagree” to “strongly agree”. The NIH-HEALS is underpinned by these multi-dimensional aspects of healing and given the evidence that it has acceptable face and content validity properties, [30] it can be used to inform the integration of psychological/spiritual healing into cancer care to stimulate the development of holistic cancer care services which align with patient perspectives of healing [16].

### 2.1. Recruitment and Data Collection

The study included adult patients (18 years and above), had sufficient cognitive ability to give informed consent, were able to communicate in the two dominant language dialects of Luganda and English, and were living with advanced cancer, receiving care at Hospice Africa Uganda in Kampala. Patients were consecutively recruited from the outpatient clinic, homebased care, or community outreaches if they met the inclusion criteria. The interviewers were introduced to potential study participants by the facility clinical teams, in a face-to-face interaction. Interviews were conducted in quiet locales, (in the patient counselling rooms, under tree sheds or sitting rooms) and were audio recorded. In addition, they also answered the open-ended question “what does the term “healing” mean to you”. These textual data were collected to generate more insights into the conceptualization of the “healing” construct from patients’ perspectives. 

The data were collected by two interviewers (EN and LM). LM is a theologian, philosopher, and psychologist and EN is a palliative care specialist with training in clinical psychology and spirituality. A total of thirty-five (35) adult cancer patients were recruited. All interviews were transcribed verbatim, those that were conducted in local languages were translated into English and thereafter the transcripts were reviewed against the audio recording by an expert in African languages, who was also fluent in the respective language. Disagreements were resolved through discussion. Ethical approval to conduct the primary study was obtained from the Hospice Africa Uganda Research and Ethics Committee and the protocol was registered with Uganda National Council for Science and Technology (HS957ES). The parent study protocol also received approval from the National Institutes of Health. All patients gave written informed consent and were free to choose to be audio recorded or not. 

This secondary analysis is focused on the patient-reported meaning for the term: “healing”. Prior to undertaking the secondary data analysis, we assessed the content of the responses were reviewed against the planned research question to establish the relevance of pre-existing qualitative data to this secondary analysis, the general quality of data, and the trustworthiness of the original data base [31]. The secondary analysis was conducted by the parent study investigators, which made it easy to assess the fit between the secondary research question and the primary dataset. 

### 2.2. Data Management and Analysis

The textual responses relating to “what does the term healing mean to you” were typed out in pre-designed word templates and thereafter imported into NVivo 12 for analysis. Data were analyzed in line with interpretive phenomenological analysis methods [32]. The analysis was done in two phases. Phase 1 aimed to identify key themes in phase 2 we clustered themes against domains of health as defined by the WHO (i.e., physical, pyscho-social, and spiritual).

For phase 1 Data were analyzed thematically following the standard six steps (i) Data familiarization; (ii) Generating initial codes; (iii) Searching for themes, (iv) reviewing themes, (v) Defining and naming themes [33]. As part of the familiarization process, EN (author and member of the study team) transcribed the data related to the definitions. She then reviewed the responses to generate initial codes, which were shared with a clinical psychologist (RB) and the two then developed themes. 

For phase 2—We listed all the responses to the question “what does the term healing means to you” (i.e., horizonalization). Each response was then assessed by two reviewers to establish if it contains a description of an experience, those that did not meet these criteria were dropped. The textual data were then clustered by thematic area by the lead researcher and were double-checked by an independent clinical psychologist and another member of the team. We then reduced the textual data to establish connections between themes, and clustering them as appropriate [34]. Major themes were common to all participants, while themes and sub-themes were created from fewer patient extracts [34]. The clustered themes were then used to develop textual descriptions of the experience and the structural descriptions were subsequently developed. The textual and structural descriptions were then integrated into the meanings and essences of the meaning of the phenomenon of interest i.e., healing [32]. At this point, we also recorded comments and pointers of potential inference for meaning as well as convergences and contradictions. The themes were then clustered by core domains of health and well-being which are: physical for example, symptoms and functional status; spiritual for example beliefs, meaning, and religion; psychological for example, cognition and emotions; social and cultural for example, family and friends, and financial) [35].

## 3. Results

### Socio-Demographic Characteristics

Thirty-five patients participated in the parent study and answered the question on meaning of healing, which informs this secondary analysis. The median age was fifty-six years, range 21–86 years and about half were male 18 (51%). The majority resided in rural settings (67%), 77% reported having a primary caregiver and about half (51%) attained primary education. Ten (28.6%) of the thirty-five participants had cancer of the prostate and (28.6%) had cancer of the cervix (see Table 1 for details). 

We identified ten major themes namely, acceptance, surrender, faith, hope, peace, freedom from suffering (for example pain, problems, or anything that bothers them), overcoming, or transcending disease, positive emotions (for example happiness), and recovery. Two participants discussed healing synonymous with death and social relations and social support. In the subsequent section we discuss each of these themes.

**Acceptance** was one of the themes that emerged, and this was described as patients having to accept the situation they are in and coming to terms with it.


*… first accept your situation and acceptance helps one to heal.*
**UG-21-2021**


**Surrender** was another theme, described as appreciating the strength within a higher power, the willingness to surrender the overwhelming circumstances to this higher power, and potentially believing that this higher power is in control.


*Accept God. Surrender your life to God. Trust him completely. Accept his will.*
**UG-34-2021**


Regarding the theme of **faith**, patients described healing believing in a higher power going to or having confidence in the higher power to heal them. This was commonly linked to God.

*Spiritual healing is very good and that means coming to God and believing that God is in control of everything instead of wandering. It would be good for everyone to come to God for spiritual healing*. **UG-05-2020**

Patients also described healing in the realms of **hope** (i.e., living with hope and hoping for something). For example, hoping that all will be well despite death being imminent or being unsure of their survival or having hope that God will in one way or the other cure the patient of their illness.

In addition, patients described **peace** in relation to “being” and “feeling” at peace and they often linked this to having God’s blessings.

*Have God’s blessing to feel peace and to heal. God can take away the cancer*. UG-33-2021

In the theme of freedom from suffering, patients described healing as **freedom from all sorts of pains**, and things that bother them or problems of the world.

*To be free from all pains.* UG-02-2020

*To free of the problems of the world.* UG-03-2020

Others described healing as **cure**, triumph against overcoming being able to transcend the physical illness. 

*It means I am cured from the sickness; it no longer affects me.* UG-08-2020

*Healing to me means you have been having a problem affecting you which you have to overcome in one way or the other. You may have the means to do so, you may not have the means to do so. If achieved, it should be able to solve that difficulty which you have say some condition of health, some difficulty in life, some challenges which are about to demolish your life, you need something to help you out of it.* UG-35-2021

Interestingly, to some **transcending** the illness or disease meant **death** as also explained by one of the respondents who noted that,

Others may be critically sick and pray that God takes them, and they call that healing. UG-15-2021 

*It means death*- UG-32-2021

Some patients linked healing to **positive emotions**, such as happiness and enjoyment of moments of life

*Healing from sickness means happiness.* UG-09-2020

Healing was also described as cure or recovery from disease, sickness or distress.

*It means getting totally cured of the disease.* UG-29-2021

Another participant described appreciating the value of having quality social networks and social support who help them accept the things they are not able to change, are a source of information and also provide company so that the patients are not isolated.

*My friends have been valuable they have taught me much about threats to life, seeking care and accepting things you cannot do. Relating to friends have been very valuable to me. So, I am not alone.* UG-34-2021

Other details are presented in Table 2.

These themes were then further clustered into broader domains based on the domains of health which include the spiritual, psychological, social and physical. 

## 4. Discussion

This study aimed to explore the meaning of the term healing from the perspective of adult patients with advanced cancer. The themes identified suggested a multi-dimensional definition framed under the following themes: acceptance, surrender, hope, faith, peace, freedom from suffering (for example pain, problems, or anything that bothers them), transcending disease, happiness, recovering from disease or illness and social relations and social support. 

These findings advance our understanding of inner patient experiences and priorities regarding healing and therefore advance the evidence base for the perspectives of cancer patients on healing. Importantly, the data provide key guidance for hospice and cancer care provider to deliver person-centred and multi-dimensional care which meets the needs of patients and the families. The dominance of the psychological/spiritual domain of healing is notable and challenges the traditional approach of restricting the definition of healing to curing physical disease or recovering from disease. This definition separates healing from the soul despite the interconnectedness of the two and is thus not an ideal definition [8]. 

Patients shared several aspects that shape the healing experience, and acceptance which a part of the spirituality construct is one of them. Through acceptance, individuals come to terms with the changes, which may be irreversible and the reality that life may soon come to an end. Those that accept the reality and embrace the new normalcy, may move into a new paradigm of life of normalcy as opposed to living in denial. Acceptance reduces psychosocial distress and improves patient well-being [36]. For some patients, surrendering to a higher power or larger than self reduces their level the distress, and this may help improve their social, spiritual and psychological well-being experiences. Patients who remain in denial commonly may experience more spiritual/existential distress or pain making and may be more complex to manage [37]. The findings of this study further strengthen prior evidence on the importance of the spiritual/existential domain in advanced disease {Puchalski, 2008 #685}. The call to action is to prioritize developing best practices for integrating spiritual care into routine care and services.

Hope is another important aspect of the healing experience. In our study two main dimensions of hope emerged: living with hope and hoping for something. These two strands are in line with existing evidence as described in a systematic review which aimed to determine the current status of research on hope in palliative care and noted that the two main themes of hope are inseparable [38]. Despite living with life limiting /life threatening illnesses, and death being imminent, patients may still have hope. Hope is useful in life and in death and some evidence has shown that it enriches well-being and is associated with better health and well-being [39]. Hope has also been associated with a sense of wholeness and may function as a mediator in the healing process [40]. A study which examined the reframing of hope in in patients with advanced Non-Small Cell Lung Cancer found that patients coped positively if they learn to hope in aspects which affect the disease for example, the reality and condition of the disease and managing well any aspects that make it difficult to keep hope alive [41]. Again, this can be a pathway to fostering healing and alleviating suffering and distress.

In our study, patient descriptions of healing also encompassed faith mainly described as a belief or having trust in a higher power commonly referred to as God. When patients are faced with distressing events of life, casting the burden on to the supernatural being gives them peace of mind which could explain why this is part of the healing experience [30]. Associations between religious involvement and mental health, less functional disability, and less cognitive decline in aging have been demonstrated by previous studies [42]. Such pathways may explain the positive correlation between faith and positive psychological and social experiences even when patients experience traumatic events such as a terminal diagnosis. Patients also described peace as part of the healing construct, commonly descried in relation to a peaceful state of well-being, having a positive relationship with God and feeling blessed. Literature demonstrates a positive relationship between peace defined as perception of self in the world and interpersonal and social factors, feeling life is worthwhile, preparation for death, and faith [43]. Given the potential positive effects of peace and the fact that patients consider it as a key component of the healing process, we argue that it is critical to ask about peace in patients with advanced cancer. In fact peace has been recommended as a good proxy for assessing patient spiritual well-being [43].

Research in cancer populations has demonstrated that spiritual well-being (understood as meaning, peace and faith) is associated with high scores on quality of life in cancer patients and in survivors [44]. Studies in patients with advanced cancer have also demonstrated that positive religious coping was associated with higher scores in overall quality of life, existential and social well-being [20], and low spirituality has been associated with worse sexual function, physical, and mental health [19]. Religiosity, spirituality, and social support have also been associated with a positive impact on the mental health well-being of these patients. For example, a study which examined the role of trust/mistrust in God in people affected by cancer reported that valuing the religiosity and spirituality of patients in health settings positively impacted the health of the patients [45] Research conducted in terminally ill AIDS patients in Uganda suggests that the existential/spiritual well-being domain was the most important [46]. Other studies in patients with metastatic disease have also demonstrated that this domain has potential for growth even in the face of advancing disease and declining function [47], and this may explain the importance of patient’s attachment to it.

Freedom from pain and other forms of suffering is also highlighted as an important theme and this has been emphasized in palliative care, with an emphasis on the need to focus on both physical and non-physical pain. This finding provides evidence of the interconnectedness between the body and mind, and the social networks/family, all of which are integral to the concept of total pain as coined by Cicely Saunders [48]. This reinforces a need to undertake a holistic approach to pain assessment, taking care of the physical, social, and spiritual pain respecting the reality of pain being what the patient says it is. Spiritual/existential distress as less researched in Africa, as it is still poorly assessed and addressed, for example a study conducted among 285 patients with advanced cancer or HIV in Uganda reported 21-58% scored poorly on spiritual well-being [49]. The full integration of spiritual/existential care into palliative care is still in its infancy but should be prioritized to improve the experiences of patients living with advanced cancer.

In regard to transcendent patients used terms such as overcoming illness and death which means changing form, overcoming a challenge or problem, or getting that supernatural support to patients to overcome a problem or challenge. Transcendent experiences of dying patients have also been associated with better grief and bereavement outcomes for affected families and peaceful deaths [50]. Given that spiritual experiences have been associated with positive patient reactions, it is plausible to recommend experience-based models of care in addition to the needs based models of care [51]. The findings are in line with previous literature on suffering and nature of opportunities, which highlight the personal experiences of transcending the suffering to being transformed to wholeness in well-being [8]. 

Happiness and social connections were another important theme in regard to healing. Patients mentioned experiences such as being able to enjoy life, just feel happy, be like other people and being able to work. Paying attention to such things in patient care and support can improve their experiences. A study conducted in Iran showed a positive relationship between happiness, general health, and life expectancy in adult cancer patients [52]. This may be explained by the patient perceived vulnerability and uncertainty, which leads to engagement in self-reflection and the potential realization that life is short [53]. Through this process, some patients find spiritual meaning in deeper connections with people, work and themselves [54]. 

The interpretation and meaning of healing are important in palliative care as it enriches the scope of how health care providers make sense of patients’ cancer journey and experiences and hence improve the way we deliver care. Clarity regarding the meaning of healing is also central to developing person-centered models of care, these models which are more responsive to patient and family priorities and would optimize outcomes of care if implemented. The findings of this study provide evidence that the end-of-life experience goes beyond the physical experiences or even physical cure or recovery, to encompass spiritual and psychosocial aspects and failure to addresses these would result into unnecessary suffering [8]. Incorporating patient perspectives would help services design care packages that are aligned to patient needs and priorities which reduces avoidable suffering and improves the well-being of patients with advanced cancer and their families.

There are several limitations to the study, the study team members involved in the analysis are palliative care professionals and this could introduce some level of bias in the interpretation of meanings. This bias was, however, mitigated by having an independent clinical psychologist review the themes, clustering of themes against core domains of health, and meanings derived for the textual data. This study also focused on patients with advanced cancer, and it might be useful to explore the meaning of healing in newly diagnosed cancer patients and those in early disease stages. The analysis was based on a single question which generated additional information on the concept of healing, but the responses were quite brief, and we could not effectively interpret some of the experiences. A larger qualitative study generating more detailed information is therefore warranted. More so, four of the responses did not provide useful insights into the meaning of the term healing as intended as such the input from these four patients could not be used. Examining their socio-demographic characteristics, we observed that they were not so different from the rest of the sample in terms of age, disease stage and gender. The omission of their responses therefore does not greatly affect the central conclusions of the study.

## 5. Conclusions

The results of this study highlight the multi-dimensional nature of the healing process which should help health workers better understand the priorities of patients with advanced disease and their caregivers. The spiritual aspect of healing is a dominant theme calling for spiritual oriented holistic care, which aligns with patient needs and priorities as such, health workers should be trained and equipped with the relevant skills to deliver care that reflects these multi-dimensional priorities of symptoms and concerns.

## Figures and Tables

**Table 1 ijerph-20-01474-t001:** Socio-demographic characteristics of the study participants.

Variable	N (%)/Median and Range
Age in years	Median 56; Range (21–86)
*Interview setting*	
Home	05 (14.3%)
Health facility	25 (71.4%)
Community outreach	05 (14.3%)
*Sex*	
Male	18 (51.4%)
Female	17 (48.6%)
*Marital status*	
Married	12 (34.3%)
Widowed	11 (31.4%)
Separated/divorced	7 (20%)
Single	5 (14.3%)
*Type of residence*	
Rural	23 (65.7%)
Urban	12 (34.3%)
*Religious affiliation*	
Catholic	14 (40.0%)
Anglican	12 (34.3%)
Born-again	04 (11.4%)
Muslim	04 (11.4%)
Seventh day	01 (2.9%)
*Highest level of education*	
Primary	18 (51.4%)
Secondary	13 (37.1%)
Degree	04 (11.4%)
*Type of cancer*	
Cervix	10 (28.6%)
Prostate	10 (28.6%)
Breast	06 (17.1%)
Kaposi’s sarcoma	03 (8.6%)
Myeloma	02 (5.7%)
Leukemia	02 (5.7%)
Head and neck	01 (2.8%)

**Table 2 ijerph-20-01474-t002:** Mapping of the patient descriptions of healing on to the multi-dimensional domains of Health (*n* = 31 *).

Patient ID	Quote	Spiritual	Psychological	Social	Physical
UG-30-2021	Accept God, surrender your life to God. Trust him completely, accept his will.	Acceptance and surrender			
UG-07-2020	There is when your sickness cannot be cured but your heart accepts it.	Acceptance and surrender			
UG-14-2021	There is healing when the disease affecting you is cured say you get cough, malaria and you treat; There is when you have a condition that will not heal. So, then that is eternity.	Acceptance and surrender			Recovering from sickness or disease
UG-21-2021	Someone to heal you first accept your situation and acceptance helps one to heal.	Acceptance and surrender			
UG-28-2021	I have accepted my condition whether I will cure or not, I live with it.	Acceptance and surrender			
UG-34-2021	Accept God. Surrender your life to God. Trust him completely. Accept his will; My friends have been valuable they have taught me much about threats to life, seeking care and accepting things you cannot do. Relating to friends have been very valuable to me. So, I am not alone.	Acceptance and surrender		Social relations	
UG-02-2021	Do not go to hell, have God’s blessings to feel peace and to heal. God can take away cancer	Peace			
UG-01-2020	It means having peace.	Peace			
UG-33-2021	Have God’s blessing to feel peace and to heal. God can take away the cancer.	Peace			
UG-06-2020	It means having hope.	Hope			
UG-12-2021	Healing means you have hope to lead your life to the future	Hope			
UG-13-2021	My healing will happen, but I cannot take back this thing. I hope to get healed but the disability on my body will remain.	Hope			
UG-05-2020	Spiritual healing is very good and that is coming to God and believing God is in control of everything instead of wandering. It would be good for everyone to come to God for spiritual healing.	Faith			
UG-11-2020	believing God is going to do great things I am not the way I was.	Faith			
UG-15-2021	Healing is when I am cured of this condition. Others may be critically sick and pray that God takes them, and they call that healing. In my belief, I ask God to heal me of this sickness.	Faith			Recovering from sickness or disease
UG-25-2021	It is about believing that a person can get healed.	Faith			
UG-07-2021	It means change of illness status, believe in that healing to know that all will be well even when death is imminent.	Faith			
UG-08-2020	It means I am cured from the sickness; it no longer affects me.				Recovering from sickness or disease
UG-32-2021	It means death	Transcend			
UG-35-2021	Healing to me means you have been having a problem affecting you which you have to overcome in one way or the other; You may have the means to do so, you may not have the means to do so. If achieved, it should be able to solve that difficulty which you have say some condition of health, some difficulty in life, some challenges which are about to demolish your life, you need something to help you out of it.	Transcend			
UG-09-2020	Healing from sickness means happiness.		Positive emotions-Happiness		
UG-16-2021;	It means being in good health like others and be able to work. When you are well, you enjoy your life.		Positive emotions–wellness and enjoy life	work	
UG-18-2021;	Healing means to recover from whatever illness or distress that one may be experiencing.				Recovering from sickness or disease
UG-22-2021	It means coming out of a problem.		Recovering from a problem	Recovering from a problem	Recovering from sickness or disease
UG-23-2021	Healing is being well in my life.		Wellness		
UG-26-2021	It means getting cured of a sickness.				Recovering from sickness or disease
UG-27-2021;	Healing for me it means recovering from sickness or a problem or whatever challenge you have.		Recovering from a problem	Recovering from a problem	Recovering from sickness or disease
UG-29-2021	It means getting totally cured of the disease.				Recovering from sickness or disease
UG-02-2020	To be free from all pains.	Freedom from suffering			
UG-03-2020	To free of the problems of the world.	Freedom from suffering			
UG-09-2021	Being free in life without anything bothering your life.	Freedom from suffering			

* Four missing values—the responses did not provide useful information in relation to the question asked.

## Data Availability

All materials used for the study are available from the corresponding author upon reasonable request. The data are not publicly available due to privacy restrictions.

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
