# Peer review of "The Meaning of Healing to Adult Patients with Advanced Cancer"

_ijerph, 2023, doi:10.3390/ijerph20021474_

Round 1

Reviewer 1 Report

In response to my previous comment (why and how this is phenomenological; no description about phenomenology in the first draft), the author added a section "3. Theoretical Perspective" (line 96-109). Although the author mentions the study is Heideggarian in this short section, nowhere in the body shows Heideggarian concepts/ideas/approach. Heideggegarian's approach is very specific. The author seems to have identified hermeneutic with Heideggegarian but the former is far broader than the latter. I do not see any application of Heidegger's concepts in the body. 

I recommend deleting "phenomenological study" from the title and  "3. Theoretical Perspective" (line 96-109). Without this section and "phenomenological study," the article can convey the research findings sufficiently. 

Having this section and the subtitle makes the article unbalanced and disappoints readers who expect Heideggarian approach in the body of writing.  

Author Response

Response to Reviewer 1 Comments

In response to my previous comment (why and how this is phenomenological; no description about phenomenology in the first draft), the author added a section "3. Theoretical Perspective" (line 96-109). Although the author mentions the study is Heideggarian in this short section, nowhere in the body shows Heideggarian concepts/ideas/approach. Heideggegarian's approach is very specific. The author seems to have identified hermeneutic with Heideggegarian but the former is far broader than the latter. I do not see any application of Heidegger's concepts in the body. 

I recommend deleting "phenomenological study" from the title and  "3. Theoretical Perspective" (line 96-109). Without this section and "phenomenological study," the article can convey the research findings sufficiently. 

Having this section and the subtitle makes the article unbalanced and disappoints readers who expect Heideggarian approach in the body of writing

Thank you for your guidance on this. We have edited the title of the manuscript to remove the phrase

phenomenological study" from the phrase and the paragraph with the Theoretical Perspective sub-title.

This manuscript is a resubmission of an earlier submission. The following is a list of the peer review reports and author responses from that submission.

Round 1

Reviewer 1 Report

The article is a worthy endeavor. I find this study itself valuable. A few points need further consideration.

1. Fix typos. Line 55, 63, 211.

2. The subtitle reads "phenomenological study." In the body, there is no description/explanation about why and how it is a "phenomenological study." I suggest either:

A. Articulate what the authors mean by "phenomenological" in the intro., main body, and conclusion.

or

B. Delete "phenomenological study" from the title. One idea is replacing it to "in Uganda" or so.

3. In the body of writing, "spiritual" and "psychological" are used synonymously. In the table, however, they are separated. Most items are classified under "spiritual" instead of "psychological." Some of them can be classified under "psychological." The distinction between "spiritual" and "psychological" is ambiguous. If the author does not clearly distinguish them, revise the table. (ex. spiritual/psychological instead of two separate categories).

Reviewer 2 Report

This study represents an interesting approach to the patient's view of the meaning of healing and the different aspects on which healing should focus. The findings may facilitate the development of new holistic approaches that improve the quality of life of cancer patients in palliative care.

Introduction

I believe that the authors have done a good job of synthesis to try to theoretically justify the study. However, I believe that the introduction is short and needs to be expanded to improve the understanding of the theoretical background and the justification of the study through these.

First, the abstract mentions as an objective "to explore the meaning of spiritual and psychological healing". In contrast, the final lines of the introduction do not specify whether the meaning of the healing to be examined is spiritual, psychological or of any other kind. I believe this should be clarified and the objective should coincide in both cases.

On the other hand, I think that if you are going to focus on the spiritual and psychological, you should expose the relationship between cancer and these aspects. I do not think it is adequate to mention only that cancer is associated with distressing symptoms and concerns that negatively impact the quality of life of the patient and families. This should be further developed. The psychological impact of cancer is reflected in higher levels of depression, anxiety, stress, distress... And of course also in the quality of life and emotional well-being. I suggest that further work be done in this regard in order to theoretically justify the study.

The same applies to the spiritual aspect. It is necessary to explain how cancer and spirituality are related. A key aspect, with healing in mind, may be the spiritual and/or religious coping of the oncological disease, which is related to better health outcomes in these patients. For this purpose, it may be useful to cite a study by Pargament, who used the term religious coping to define this term: https://doi.org/10.2307/1388152. Similarly, research has amply demonstrated the positive impact of religion and spirituality on the health of cancer patients, represented by higher levels of well-being and quality of life and lower levels of anxiety, depression, or social isolation. Some reference studies may include the following:

DOI: 10.1089/jpm.2006.9.646

DOI: 10.1002/pon.929

DOI: 10.1007/s10943-019-00907-6

DOI: 10.1007/s10943-017-0468-z

One could also mention the impact that the diagnosis has on the patients' own spirituality. Religious/spiritual people may experience changes in their religiosity and/or spirituality depending on how they interpret this situation: a punishment from God, a reason to fight, etc.

Likewise, if the multidimensional impact of cancer and the need for holistic treatments are to be mentioned later, I believe it is necessary to also delve into the impact this disease has on physical (DOI: 10.1002/cncr.29353) and social aspects (DOI: 10.1080/07347332.2013.798761).

Once it is understood that cancer has a physical, psychological, social and spiritual impact, it makes more sense to talk about palliative care professionals pointing out the importance of holistic treatments focusing on all these aspects.

The final part of the introduction that focuses on healing is much more appropriate. I suggest rewording the sentences in lines 53-55. They are separated by a dot and, in this way, the second one remains loose, meaningless. I think it would be enough to change the dot for a comma. I also suggest defining the spiritual well-being mentioned in line 56 and its relationship to cancer. Here is a reference study on this subject: 

DOI: 10.1207/S15324796ABM2401_06

In addition, I suggest the authors take a look at this paper on healing and spirituality, it may be helpful for them to expand this part:  DOI: 10.21037/apm.2017.05.01

Materials and method

The method is adequate and is fully and clearly explained. I make some suggestions:

In the recruitment section, I suggest mentioning the inclusion criteria. Although they appear in the primary study, they may be useful to the reader.

In addition, the sample should be described briefly in this section, although it will be described in more depth in the results section. However, at a minimum, the number of participants and their age and/or gender should be shown.

Although the method is well understood, I would like to propose to the authors that they reorganize the information into four subsections:

1.     A subsection on participants, describing the sample and explaining how it was obtained (sampling).

2.       A subsection on instruments, explaining the NIH-HEALS.

3.       A subsection on procedure.

4.       Finally, a subsection on data analysis, as it is in the manuscript.

Results

The results are well presented so that they are clearly understood. In addition, the table helps to understand them in a more visual way.

Discussion

I consider that the findings are well discussed, but I believe that it would be necessary to broaden the discussion a little, as was the case with the introduction.

The findings have clinical implications that should be highlighted. I believe that emphasis should be placed on how patients' disclosure of their psychosocial and, above all, spiritual needs affects the practice of health care. One can mention, for example, the need to train healthcare professionals in skills that will enable them to deal with the spiritual needs of patients.

Examples could also be given of studies that have shown the benefits of holistic approaches for oncology patients.

I believe it is necessary to comment on each of the main issues identified. Therein lies the true potential of the work, not only in generalizing the need to consider the psychosocial and spiritual aspects of cancer patients. For example, faith and peace have emerged as important aspects for these people. Both constructs are part of spiritual well-being, which in turn is related to better health of cancer patients in numerous research studies. Here are a couple of examples:

DOI: 10.1017/S1478951519000464

DOI: 10.1007/s10943-020-01147-9

I believe that commenting on specific aspects such as this can add value to the work.

Another example could be happiness or social relationships. Some research has related religiosity, spirituality and social support with the emotions of these patients, observing a positive impact of these variables on the mental health of oncology patients. Here is an example: DOI: 10.3390/healthcare10061138. I consider it appropriate to rely on specific previous findings to give relevance to the findings of this study.

In short, further discussion of the results would give greater clinical relevance to the findings of this study.

Likewise, I believe it would be necessary to comment on the limitations of the study, if any.

Finally, the citation format of line 215 should be revised.